# A Novel Broadband Monopole Antenna with T-Slot, CB-CPW, Parasitic Stripe and Heart-Shaped Slice for 5G Applications

**DOI:** 10.3390/s20247002

**Published:** 2020-12-08

**Authors:** Zhendong Ding, Hao Wang, Shifei Tao, Dan Zhang, Chunyu Ma, Yunxian Zhong

**Affiliations:** 1School of Electronic and Optical Engineering, Nanjing University of Science and Technology, Nanjing 210094, China; d776128747@139.com (Z.D.); haowang@mail.njust.edu.cn (H.W.); s.tao@njust.edu.cn (S.T.); 2College of Information Science and Technology, Nanjing Forestry University, Nanjing 210037, China; Machunyu2021@163.com (C.M.); sly_zhong@163.com (Y.Z.); 3State Key Laboratory of Millimeter Waves, Southeast University, Nanjing 210096, China

**Keywords:** parasitic patch, return loss, gain, efficiency, CPW

## Abstract

This paper presents a novel broadband monopole antenna that was equipped with a bottom semicircle ground structure, a parasitic patch, a T-shaped slot, s transmission line, a parasitic strip, heart-shaped slices and a coplanar waveguide (CPW). The simulation results revealed that the proposed design had a relatively high return loss, a wide bandwidth and high efficiency. A prototype of the proposed antenna with an overall size of 0.94 λ_0_ × 0.94 λ_0_ × 0.02 λ_0_ (λ_0_ is the free-space wavelength) was fabricated and measured. The measurement results showed that the prototype had a bandwidth of 4.02 GHz (4.69–8.71 GHz) and a relative bandwidth of 60%. Besides, the maximum gain was 3.31 dBi and the maximum efficiency was 91.1% in the range of 5 to 8.5 GHz. Furthermore, it was found that the prototype almost achieved omnidirectional radiation. Its operating frequency band covered those of industrial scientific medical (ISM) (5.725–5.850 GHz), the radio frequency identification (RFID) (5.8 GHz) and the wireless local area network (WLAN) (5.15–5.25 GHz and 5.725–5.825 GHz).

## 1. Introduction

In recent years, modern wireless communication system [1] technology has developed rapidly. Since it was proposed, a series of wireless communication devices, such as mobile phones, computers, navigators, satellites and radar systems, have emerged, which shorten the distance between people, bring some fun and change people’s lifestyles. Antennas with a function of receiving and transmitting the required signal are an indispensable part of such technology. Communication technology has undergone a development process from 2G to 3G and 4G, and to 5G systems today. In the second and third communication technology systems, the size of traditional antennas proposed by Wang, Y. et al. were relatively large, its bandwidth were relatively narrow and its efficiency were low [2,3]. There are some limitations in the application. Its performance will affect the rectification system. However, further exploration is still required to develop more advanced technology, because the existing wireless communication technology is unable to meet the new needs of society with the progress rate of the times. From the global perspective, the rapid development of 4G has become a reality, 5G research is also growing rapidly and maturing, and 6G technology is being studied by scholars. Newer 5G and 6G antennas [4,5] are being researched. Antennas with multiple functions, multiple adaptability and multiple purposes are warranted. Research in the communication equipment field focuses on the miniaturization, broadband and polarization of antennas.

Although research on antennas is mature in the wireless communication system, we have still made some breakthroughs. Broadband is the most common index employed in antenna research. Its research value has been proven in many research papers. In this paper, indexes including broadband, polarization and gain were frequently used. A meta-surface (MTS) antenna with a broad bandwidth of 4.88 GHz fed by an aperture coupling through a coplanar waveguide (CPW) was proposed by Wang, J. et al. in [6]. Some articles discussed not only broadband, but also polarization, such as the antenna with a bandwidth of 4.65 GHz and circular polarization in [7], and the antenna with horizontal polarization and a bandwidth of 39 to 50 MHz in [8]. Antennas with high gain were also studied in [9]. Some researchers tried to minimize the size of an antenna to achieve miniaturization without affecting its performance [10,11]. Methods to improve the bandwidth of microstrip antennas have been investigated in many papers. The monopole structure in [12], T-shaped slots in [13], CPW in [14], parasitic strips in [15,16], substrate integrated waveguide (SIW) in [17] and parasitic patches in [18] were demonstrated to expand the antenna bandwidth. However, the implementation method of the above structure is relatively simple and further research is needed to combine various methods to achieve broadband function.

In practice, the frequency band of antennas should be designed according to their engineering applications. The frequency band of the novel antenna designed in this paper is applicable to IEEE (Institute of Electrical and Electronics Engineers) 802.11a [19,20], the WLAN in 5.15–5.25/5.725–5.825 GHz and IEEE 802.16 [21,22] and the WiMAX system in 2.5–2.69/3.4–3.69/5.25–5.85 GHz. Figure 1 shows the explosive decomposition structure of the designed antenna. It can be seen from this figure that the whole antenna comprises a monopole structure, a parasitic patch, a parasitic strip, a conductor-backed coplanar waveguide (CB-CPW), a T-shaped slot, heart-shaped slices and a back semicircle ground structure. The CPW structure and the semicircle ground plate on the back constitute the conductor-backed CB-CPW.

In this paper, a novel broadband monopole antenna with a geometric dimension of 38 × 38 × 1 mm^2^ was designed and fabricated. In order to expand the bandwidth and reduce the size of the antenna, many elements were incorporated into the structure, including a semicircle ground, a CPW, a parasitic patch, a parasitic strip, a T-shaped slot and a heart-shaped slice. Irregular gaps can produce a stronger resonance effect and wider bandwidth. Symmetrical structures are coupled with each other and resonate with each other, which can achieve good impedance matching and realize broadband function. The antenna can be used in 5G communication fields, providing a wide bandwidth for wireless communication technology. The proposed antenna achieved miniaturization, a wide band and horizontal polarization. This paper systematically describes the antenna design in this paper, which is composed of the introduction, the analysis and design of the antenna, simulation and measurement, a discussion and a conclusion. This paper, from theory to practice, finally achieves the desired effect.

## 2. Analysis and Design of the Antenna

### 2.1. Antenna Analysis

Impedance matching is important in the design of antennas. Antennas that achieve impedance matching can transmit power efficiently. However, impedance mismatching leads to the generation of various unnecessary reflection signals and recoil signals. The impedance matching quality of antennas is usually measured by voltage standing wave ratio (VSWR), reflection coefficient (S11) and other parameters. The gain, efficiency and input impedance of the antenna are also critical factors influencing the performance of antennas. During the designing process, a parasitic strip and heart-shaped slices were added to the antenna to acquire the effect of resonance.

CB-CPW is composed of semicircle ground and CPW, which belongs to parallel plate mode. The coupling of CB-CPW can lead to a sharp decrease in impedance and an increase in reflection coefficient—that is, an increase in bandwidth. It has good heat dissipation effect and high mechanical strength.

The resonant frequency of the CB-CPW structure in [23] can be expressed as: (1)$$f_{mn}=\frac{c}{2\sqrt{\varepsilon_{r}}}{\left[{\left(m/w_{g}\right)}^{2}+{\left(n/l_{g}\right)}^{2}\right]}^{0.5}$$
where *ε**_r_* is the relative permittivity of the material, *c* is the speed of light, *w_g_* and *l_g_* are the width and length of the ground plane at both ends of the CB-CPW, respectively, and *m* and *n* are factors of the resonant mode.

Figure 2 shows the simplified equivalent circuit model, where $L^{\prime }\approx 0$. The input impedance of the TM*_mn_* mode is: (2)$$Z_{in}\approx \frac{1}{G_{mn}+j[\omega C_{mn}-1/(\omega L_{mn})]}$$

Equation (2) may also be rewritten as:(3)$$Z_{in}=\frac{1}{1+jQ[f/f_{mn}-f_{mn}/f]}=\frac{R_{mn}}{1+jQS}$$
(4)$$S=\frac{f}{f_{mn}}-\frac{f_{mn}}{f}=\pm \frac{\rho -1}{Q\sqrt{\rho }}$$
where *ρ* denotes the voltage standing wave ratio (VSWR).

The relative bandwidth is:(5)$$BW=\frac{f_{H}-f_{L}}{f_{mn}}\times 100\%=\frac{\rho -1}{Q\sqrt{\rho }}\times 100\%$$
(6)$$D=\frac{{\left|P(\theta ,\varphi )\right|}_{\max}}{\frac{1}{4\pi }\oint \left|P(\theta ,\varphi )\right|d\Omega }$$
(7)G=ηD
where *f_H_* is the highest frequency point and *f**_L_* is the lowest frequency point of 10 dB.

The resonant frequency of the equivalent circuit is: (8)$$f_{o}=\frac{1}{2\pi \sqrt{L_{mn}C_{mn}}}$$

The directivity coefficient *D* is the ratio of the peak energy to the average energy at a specific angle. Considering the loss of the antenna, the gain *G* is equal to the directivity coefficient multiplied by the antenna efficiency. The antenna efficiency is also the ratio of antenna radiation power to input power. If the loss of the feeder system is included, the antenna gain is called the actual gain.

### 2.2. Antenna Design

The geometry and dimensions of the proposed novel broadband monopole antenna are shown in Figure 3. The antenna was designed on a single-layer FR4 (Epoxy glass cloth laminate) substrate with a dielectric constant of 4.4 and a loss tangent of 0.02. Other units of the antenna were made of copper. This design was analyzed and studied by high frequency structure simulator (HFSS) software, and some size parameters of the antenna were optimized and compared. In the main mode, the lumped port was used to excite the transmission line and a 50-Ω load was introduced into the design.

Figure 4 shows five kinds of antennas. Antenna 1 is a common monopole T-slot antenna. Antenna 2 is prepared by adding a semicircular ground plane to Antenna 1. Antenna 3 is made by embedding a parasitic strip into Antenna 2. The structures of Antenna 4 and Antenna 5 are obtained by cutting heart-shaped slices off from Antenna 3. It can be seen from Figure 5 that after adding the parasitic strip to the CPW of Antenna 3, a bandwidth of 10 dB was achieved, and the bandwidth of Antenna 5 is larger than that of Antenna 3, indicating that heart-shaped slices could greatly enlarge the bandwidth. Parallel strip is very important to bandwidth expansion. The reason why antennas with irregular slots have a wider bandwidth is because they have stronger resonance effects. The symmetrical slots on the patch also resonated with each other. Symmetrical slots on two sides of the feed line are impedance-matched when they are coupled with the feed line, thus improving the S11.

According to Figure 5, the bandwidth of Antenna 5 is the largest, and its simulation bandwidth is 3.42 GHz (4.87–8.29 GHz). The two peak points of its bandwidth are 10.826 and 10.632 dB, which are all greater than 10 dB. Therefore, Antenna 5 excites TM_10_, TM_20_ and TM_30_ modes and realizes three resonant points.

To facilitate the cutting, three pairs of *X*-axis-symmetrical *P* points were located to make a single heart-shaped slice, as shown in Figure 6, where *P* of each heart shape corresponds to the values of x and y (x1 *=* −20, x2 *=* −16, x3 = −12, x7 *=* 4, x8 = 4, x9 *=* 10, y1 *=* −16, y2 = −4, y3 *= −*15, y7 *= −*18, y8 *=* −2, y9 *=* −10; unit: mm).

Figure 7 optimizes the parameters of the transmission line length, b, and the T-slot width, c, of the antenna. When b = 17.1 mm and c = 3.5 mm, the antenna had the widest broadband. Figure 8 modifies the radius, R, of the back circle of the antenna as well as the length, f, and width, g, of the parasitic strip. When R = 9.1 mm, f = 3.4 mm and g = 0.1 mm, the antenna achieved the optimal broadband function. The simulation results showed that the center frequency was 6.58 GHz and the maximum return loss was 39.29 dB at 5.1 GHz. When S11 ≤ −10 dB, *f**_H_* = 8.29 GHz and *f**_L_* = 4.87 GHz, the absolute bandwidth, B, and the relative bandwidth, B_r_, were *f**_H_* − *f**_L_* = 3.42 GHz and 52%, respectively.

As indicated by Figure 9, the current of the proposed antenna at three different frequencies was mostly distributed on the monopole transmission line and the feed line but rarely at the edge of the slot. There was no strong current distribution on the semicircle ground.

## 3. Simulation and Measurement

The antenna was processed and measured precisely to verify its feasibility. In the measurement process, the antenna was calibrated and the cable loss was reduced. Figure 10 illustrates the fabricated prototype with photos. The center frequency of the antenna was 6.7 GHz, the maximum call return loss was 39.89 dB at 8.33 GHz, the bandwidth was 10 dB at 4.02 GHz (Figure 11a), the VSWR was less than 2 at 5–6 GHz (Figure 11b), the gain was 0.01–3.31 dBi at 5–6 GHz (Figure 12a) and the efficiency was 79.01–90.9% at 5–6 GHz (Figure 12b). The measurement result of the standing wave was excellent and its flatness was within 1.5. The measured antenna bandwidth was larger than the simulated antenna bandwidth, and the measurement effect was obvious. It shows that the measurement and simulation (theoretical and practical) have a certain error, but the impact is small, and the measured data and simulation data of the antenna are basically consistent.

Figure 13a–f present the 2D radiation patterns (E- and H-plane) simulated and measured at 5.8, 5.5 and 5.2 GHz, respectively. According to those figures, the antenna almost achieved omnidirectional radiation. The measurement and simulation results of the designed antenna were basically consistent. 

Table 1 compares several antennas, each of which has its own advantages and innovations. According to the measurement results, the antenna has a bandwidth of 4.02 GHz (4.69–8.71 GHz), a relative bandwidth of 60%, a maximum gain of 3.31 dBi and a maximum efficiency of 91.1%. The performance of the ultra-wideband (UWB) is realized with high efficiency and suitable gain. The antenna proposed in this paper had a smaller size and a wider bandwidth.

## 4. Discussion

The results of this study suggest that a miniaturized patch antenna with extended bandwidth can be realized after parameter optimization by ANSYS HFSS (Ver. 15). This paper analyzed the return loss of the antenna to observe its bandwidth. It can be seen from the design part that when a T-slot and parasitic patch are inserted, the broadband effect is not ideal. When a parasitic strip is added, the antenna has a bandwidth of about 1 GHz. Furthermore, cutting into two heart-shaped slots does not change, and cutting into four heart-shaped slots has a broadband effect, achieving ultra-wideband performance. This kind of antenna is also based on CPW structure. Many kinds of structure combinations excite the resonance effect to obtain the required broadband antenna.

Compared with Table 1, the bandwidths of the antennas in references [24,29] are wider than that in this paper, but their gain and efficiency are lower. Similarly, the gain of the antenna in reference [25] is lower than that in this paper, and the efficiency of the antenna in reference [32] is lower than the proposed antenna. The antenna size of this paper is the smallest compared with the references in Table 1. From Figure 12, it can be seen that the gain of the antenna is relatively smooth and the antenna has strong stability. For the cross-polarized pattern, the E-plane can radiate in all directions. Generally speaking, the design of the antenna unit in this paper is in accordance with the theoretical results. If necessary, at a later stage, the antenna in this paper can continue to be expanded in this theory, and this paper can be extended to the broadband array structure in the future.

There were minor errors between the measurement and simulation (theoretical and practical) data, and the impact was small, which may be caused by several reasons. First, the resonance frequency of the processed antenna may be higher, and frequency offset leads to this kind of error. Second, the reason may be related to the HFSS software, such as the establishment of the model and the way of feeding. Third, the test cable loss, the accuracy of antenna processing, the feed welding during the test and the antenna test environment also possibly resulted in the difference between the measured and the simulated data.

## 5. Conclusions

A novel broadband monopole antenna is presented in this paper. Its VSWR bandwidth is measured to be 4.02 GHz (4.69 to 8.71 GHz), its efficiency is 73.8% to 91.1% at 5–8.5 GHz and its peak gain is 3.31 dBi. This broadband and miniaturized antenna is easy to process and assemble, with fast heat-dissipating speed, a low cost, small radiation loss, high mechanical strength and a beautiful appearance. Therefore, it has broad application prospects. It can operate in the frequency bands of ISM (5.725–5.850 GHz), RFID (5.8 GHz) and WLAN (5.15–5.25 GHz and 5.725–5.825 GHz). 

## Figures and Tables

**Figure 1 sensors-20-07002-f001:**
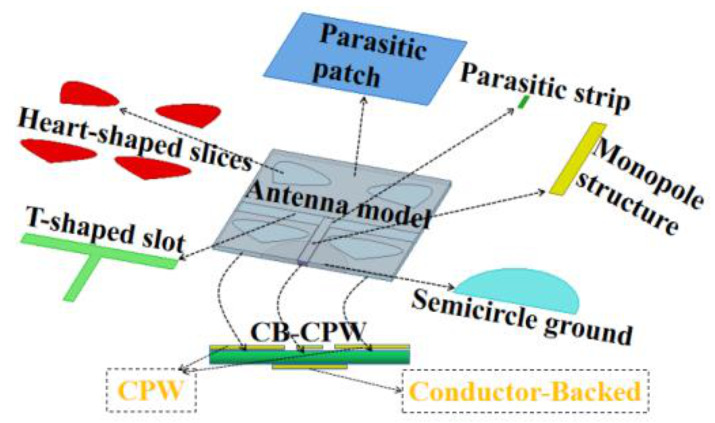
Explosive decomposition structure.

**Figure 2 sensors-20-07002-f002:**
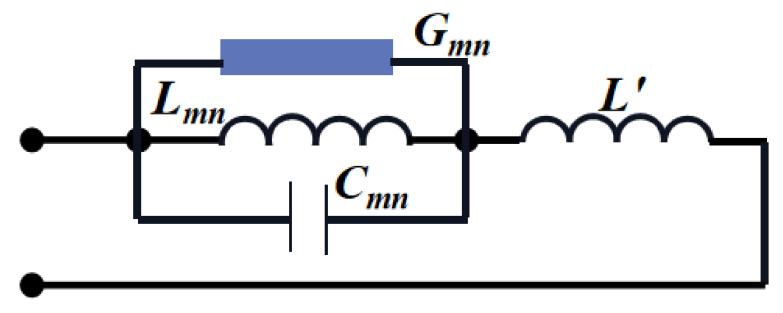
The simplified equivalent circuit model (established when *w*_mn_ is isolated from all other resonant frequencies).

**Figure 3 sensors-20-07002-f003:**
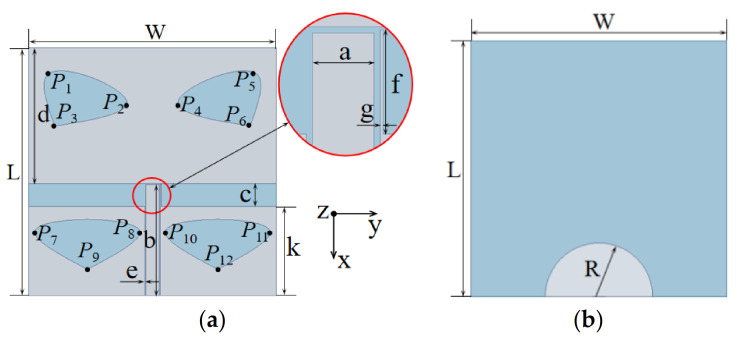
Geometric configuration of the proposed antenna: (**a**) top side and (**b**) bottom side, where L = W = 38 mm; H = 1 mm; R = 9.1 mm; a = 2 mm; b = 17.1 mm; c = 3.5 mm; d = 20.7 mm; e = 0.1 mm; f = 3.4 mm; g = 0.1 mm; k = 13.8 mm.

**Figure 4 sensors-20-07002-f004:**
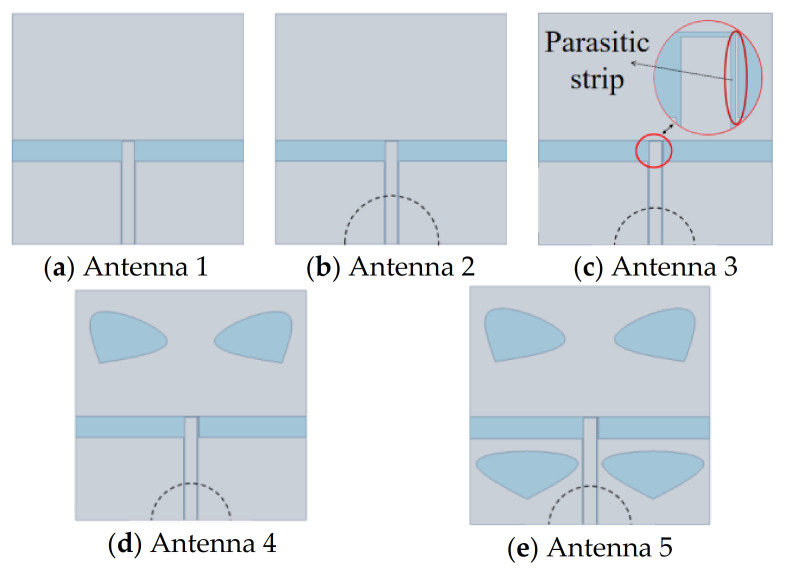
Geometry comparison of five types of antennas.

**Figure 5 sensors-20-07002-f005:**
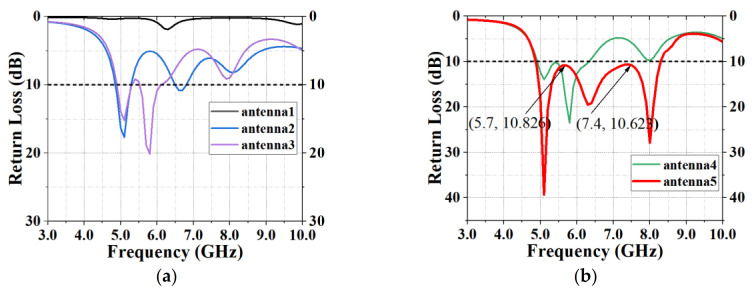
Comparison of return losses of five antennas with different shapes.

**Figure 6 sensors-20-07002-f006:**
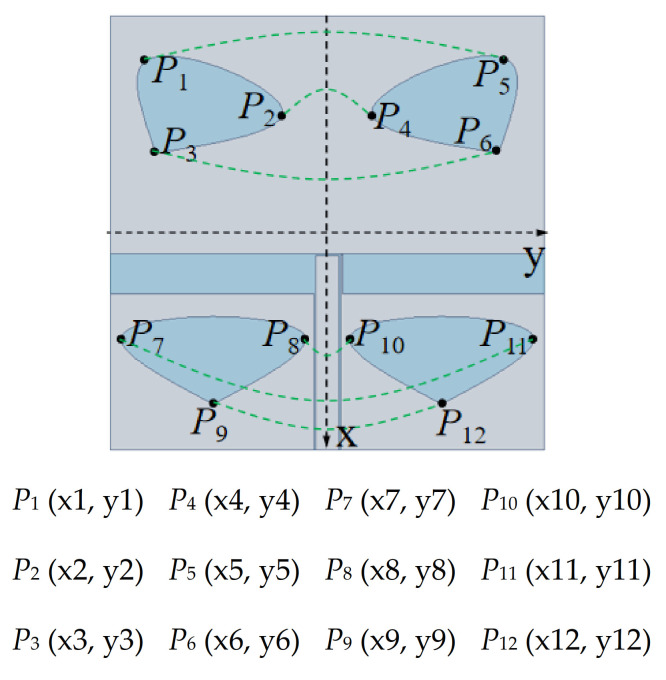
Symmetry analysis of the heart-shaped structure.

**Figure 7 sensors-20-07002-f007:**
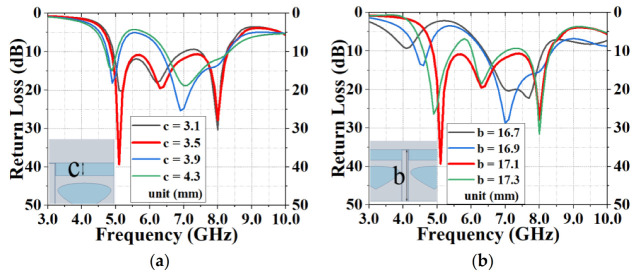
Improvement of the return loss by optimizing (**a**) transmission line length, b, and (**b**) T-slot width, c.

**Figure 8 sensors-20-07002-f008:**
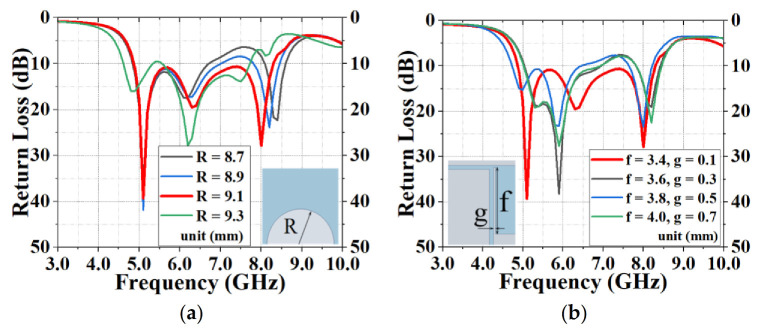
Improvement of the return loss by optimizing (**a**) the radius, R, of the back circle of the antenna and (**b**) the length, f, and width, g, of the parasitic strip.

**Figure 9 sensors-20-07002-f009:**
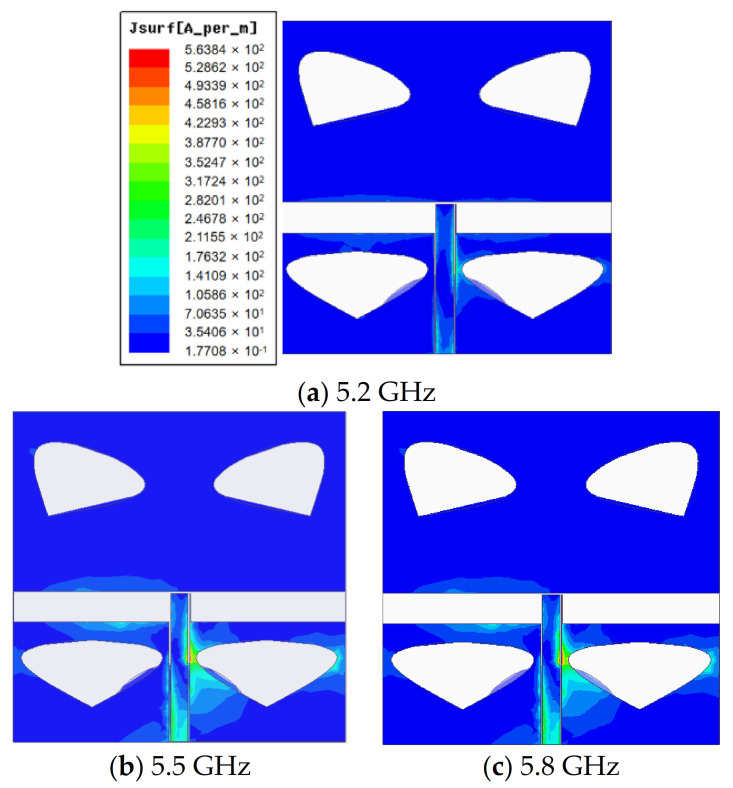
Simulated surface current distribution of the proposed novel antenna at frequencies of 5.2 (**a**), 5.5 (**b**) and 5.8 (**c**) GHz.

**Figure 10 sensors-20-07002-f010:**
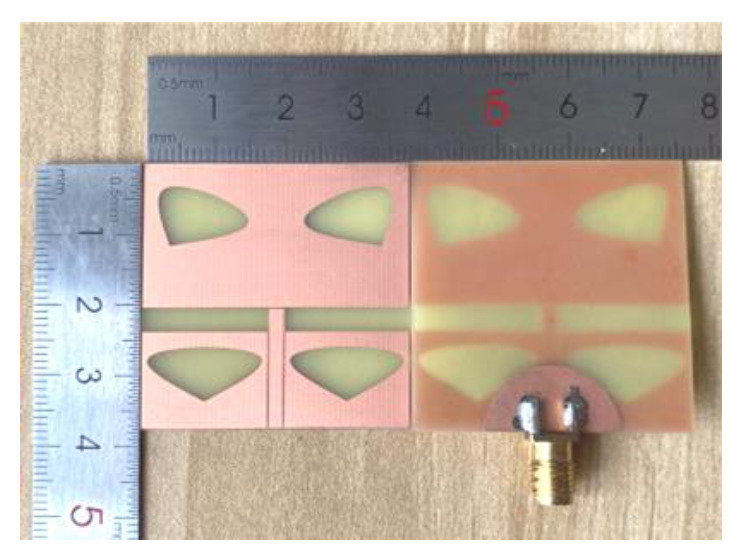
Photo of the front and the back of the designed antenna.

**Figure 11 sensors-20-07002-f011:**
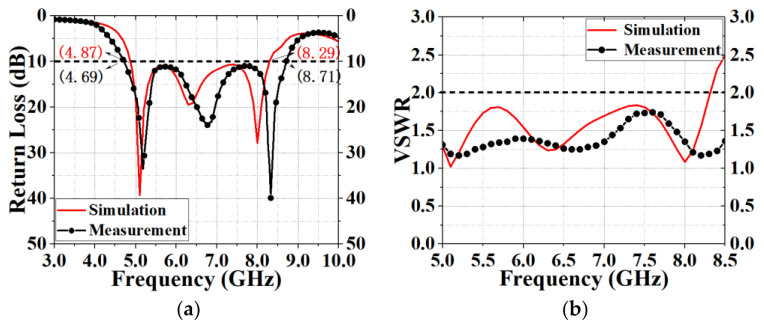
Comparison of simulated and measured (**a**) return loss and (**b**) voltage standing wave ratio (VSWR) of the designed antenna.

**Figure 12 sensors-20-07002-f012:**
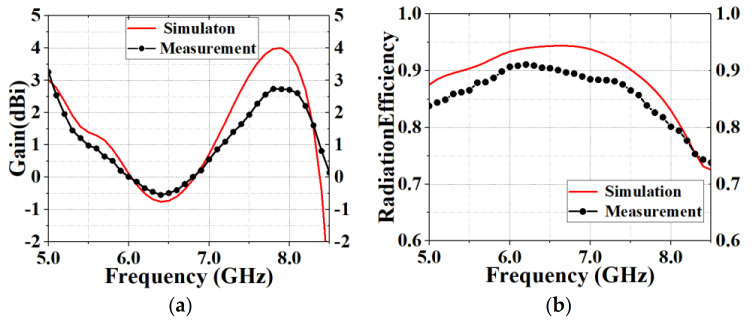
Comparison of simulated and measured (**a**) gain and (**b**) radiation efficiency of the designed antenna.

**Figure 13 sensors-20-07002-f013:**
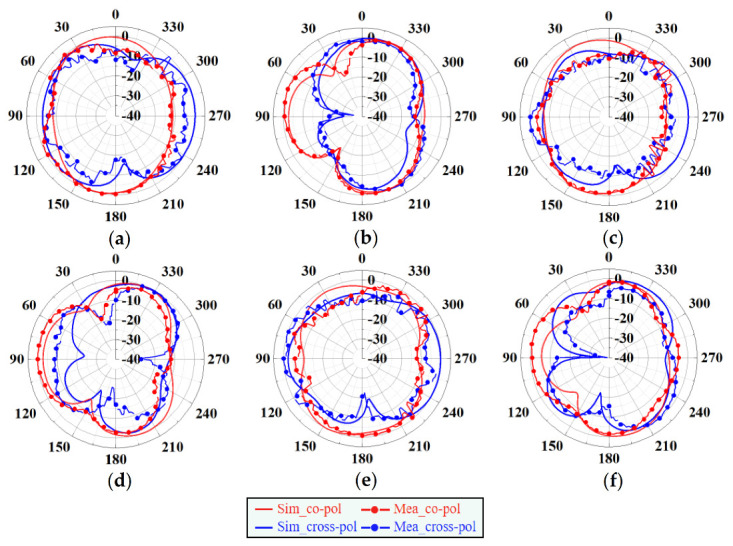
The measured and simulated 2D radiation patterns of the proposed antenna. (**a**) E-plane at 5.2 GHz. (**b**) H-plane at 5.2 GHz. (**c**) E-plane at 5.5 GHz. (**d**) H-plane at 5.5 GHz. (**e**) E-plane at 5.8 GHz. (**f**) H-plane at 5.8 GHz.

**Table 1 sensors-20-07002-t001:** Comparison between the proposed Antenna 1 and previously designed antennas.

Ref.	Size (mm^3^)	Imp. Bandwidth (GHz, %)	Max Gain (dBi)	Max Efficiency (%)
[24]	63 × 75 × 1.6	1.81–1.63, 71.63%	2.5	90%
[25]	55 × 50 × 1	1.75–4.5, 88%	2.3	95%
[26]	90 × 40 × 0.79	2.4–4.2, 54.5%	3	/
[27]	60 × 50 × 1.5	3.44–6.13, 56.2%	/	89%
[28]	40 × 40 × 1	4.83–8.17, 55%	3.87	/
[29]	48 × 43 × 0.8	0.115–2.90, 185%	2.35	78.85%
[30]	60 × 40 × 0.8	2.3–3.15, 31.2%	3.7	/
[31]	70 × 70 × 1.6	1.5–2.5, 50%	5.8	/
[32]	50 × 50 × 1	1.71–3.66, 73.3%	4	80%
Proposed	38 × 38 × 1	4.69–8.71, 60%	3.31	91.1%

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
