# Peer review of "A Novel Broadband Monopole Antenna with T-Slot, CB-CPW, Parasitic Stripe and Heart-Shaped Slice for 5G Applications"

_sensors, 2020, doi:10.3390/s20247002_

Round 1

Reviewer 1 Report

1.The authors may include the related work section. The related work section can be extended with more comprehensive and detailed
review.

2. The performance from the evaluation of the design of the Antenna is not compared to other traditional Antenna used in second and third communication technology system.

3. It is preferable the authors to provide a comparison in the design that shows the extended components from the current work so that readers can easily catch new parts compared to existing design of Antenna.

4. Minor spell check required.

Reviewer 2 Report

In this paper, the authors proposed a novel broadband monopole antenna is presented in this paper. Obviously, the novel and interesting points can be seen in this paper, and I think this paper can be published after doing some minor revision.

  • English in this paper can be considered to be improved. Some errors also can be seen in some sentences in such an article.
  • I think some figures have to be replotted again. For instance, in Fig.5, there are too many curves in one figure, which can lead to being unreadable. Maybe plotting them in two figures is a better choice.
  • As mentioned in Fig.12, in the frequency region of 6- 7GHz, the Gain is less than zero. The authors can consider giving some methods to improve the Gain in such a frequency region as a future direction of the investigation.

Reviewer 3 Report

The introduction to the paper is not well organised and does not provide a clear picture of the state of the art or about the different approaches adopted. A more chematic approach should be preferred, clearly indicating the different directions and commenting on why such direction is not fully appropriate (at least to the authors' view). The organisation of the paper should be given in the introduction too.

While describing the antenna design, the approach should be more systematic. I appreciate the simultaneous simulation of the incremental version of the proposed antenna, but no in-depth explanation is given on the reason for introducing, from one version to the other, the modification. For example, wht is the effect of the semi-circular ground ? current distribution with and without ? Most importantly, if the effect is not crucial, why authors keep it in the final antenna ? In other words, a step-by-step approach should be followed, and each one duly justified.

The above consideration partly applies also to the heart-shaped sectors introduction. In this case a minimum analysis is provided, but it is basically lost in the many constructive delays. On the ither hand, such details do not include the basic optimisation related to the distance and size of th shape(s). Even in this case a more schematic approach should be provided. All the geometrical parameters should be provided in a tabuar format, and not be introduced in the text.

The figures providing the antenna performance comparison (figs 11 and 12) are really difficult to understand, given the many nubers superimposed on the figures themselves. Please describe the comparison in a more clear way, avoiding the use of insets and try to justify the deviations from measured performance.

Some work on english gramar and spelling is to be performed.

Round 2

Reviewer 3 Report

Some minor checks still needed, regardless of the certification provided (for instance p.2, line 53, either "its" or "the" ...)